# ADVERSARIAL IMITATION VIA VARIATIONAL INVERSE REINFORCEMENT LEARNING

**Ahmed H. Qureshi**
Department of Electrical and Computer Engineering
University of California San Diego,
La Jolla, CA 92093, USA
a1qureshi@ucsd.edu

**Byron Boots**
College of Computing
Georgia Institute of Technology
Atlanta, GA 30332, USA
bboots@cc.gatech.edu

**Michael C. Yip**
Department of Electrical and Computer Engineering
University of California San Diego,
La Jolla, CA 92093, USA
yip@ucsd.edu

## ABSTRACT

We consider a problem of learning the reward and policy from expert examples under unknown dynamics. Our proposed method builds on the framework of generative adversarial networks and introduces the empowerment-regularized maximum-entropy inverse reinforcement learning to learn near-optimal rewards and policies. Empowerment-based regularization prevents the policy from overfitting to expert demonstrations, which advantageously leads to more generalized behaviors that result in learning near-optimal rewards. Our method simultaneously learns empowerment through variational information maximization along with the reward and policy under the adversarial learning formulation. We evaluate our approach on various high-dimensional complex control tasks. We also test our learned rewards in challenging transfer learning problems where training and testing environments are made to be different from each other in terms of dynamics or structure. The results show that our proposed method not only learns near-optimal rewards and policies that are matching expert behavior but also performs significantly better than state-of-the-art inverse reinforcement learning algorithms.

## 1 INTRODUCTION

Reinforcement learning (RL) has emerged as a promising tool for solving complex decision-making and control tasks from predefined high-level reward functions (Sutton et al., 1998). However, defining an optimizable reward function that inculcates the desired behavior can be challenging for many robotic applications, which include learning social-interaction skills (Qureshi et al., 2018; 2017), dexterous manipulation (Finn et al., 2016b), and autonomous driving (Kuderer et al., 2015).

Inverse reinforcement learning (IRL) (Ng et al., 2000) addresses the problem of learning reward functions from expert demonstrations, and it is often considered as a branch of imitation learning (Argall et al., 2009). The prior work in IRL includes maximum-margin (Abbeel & Ng, 2004; Ratliff et al., 2006) and maximum-entropy (Ziebart et al., 2008) formulations. Currently, maximum entropy (MaxEnt) IRL is a widely used approach towards IRL, and has been extended to use non-linear function approximators such as neural networks in scenarios with unknown dynamics by leveraging sampling-based techniques (Boularias et al., 2011; Finn et al., 2016b; Kalakrishnan et al., 2013). However, designing the IRL algorithm is usually complicated as it requires, to some extent, hand engineering such as deciding domain-specific regularizers (Finn et al., 2016b).

Rather than learning reward functions and solving the IRL problem, imitation learning (IL) learns a policy directly from expert demonstrations. Prior work addressed the IL problem through behavior cloning (BC), which learns a policy from expert trajectories using supervised learning (Pomerleau, 1991). Although BC methods are simple solutions to IL, these methods require a large amount of

data because of compounding errors induced by covariate shift (Ross et al., 2011). To overcome BC limitations, a generative adversarial imitation learning (GAIL) algorithm (Ho & Ermon, 2016) was proposed. GAIL uses the formulation of Generative Adversarial Networks (GANs) (Goodfellow et al., 2014), i.e., a generator-discriminator framework, where a generator is trained to generate expert-like trajectories while a discriminator is trained to distinguish between generated and expert trajectories. Although GAIL is highly effective and efficient framework, it does not recover transferable/portable reward functions along with the policies, thus narrowing its use cases to similar problem instances in similar environments. Reward function learning is ultimately preferable, if possible, over direct imitation learning as rewards are portable functions that represent the most basic and complete representation of agent intention, and can be re-optimized in new environments and new agents.

Reward learning is challenging as there can be many optimal policies explaining a set of demonstrations and many reward functions inducing an optimal policy (Ng et al., 2000; Ziebart et al., 2008). Recently, an adversarial inverse reinforcement learning (AIRL) framework (Fu et al., 2017), an extension of GAIL, was proposed that offers a solution to the former issue by exploiting the maximum entropy IRL method (Ziebart et al., 2008) whereas the latter issue is addressed through learning disentangled reward functions by modeling the reward as a function of state only instead of both state and action. However, AIRL fails to recover the ground truth reward when the ground truth reward is a function of both state and action. For example, the reward function in any locomotion or ambulation tasks contains a penalty term that discourages actions with large magnitudes. This need for action regularization is well known in optimal control literature and limits the use cases of a state-only reward function in most practical real-life applications. A more generalizable and useful approach would be to formulate reward as a function of both states and actions, which induces action-driven reward shaping that has been shown to play a vital role in quickly recovering the optimal policies (Ng et al., 1999).

In this paper, we propose the empowerment-regularized adversarial inverse reinforcement learning (EAIRL) algorithm[1]. Empowerment (Salge et al., 2014) is a mutual information-based theoretic measure, like state- or action-value functions, that assigns a value to a given state to quantify the extent to which an agent can influence its environment. Our method uses variational information maximization (Mohamed & Rezende, 2015) to learn empowerment in parallel to learning the reward and policy from expert data. Empowerment acts as a regularizer to policy updates to prevent overfitting the expert demonstrations, which in practice leads to learning robust rewards. Our experimentation shows that the proposed method recovers not only near-optimal policies but also recovers robust, transferable, disentangled, state-action based reward functions that are near-optimal. The results on reward learning also show that EAIRL outperforms several state-of-the-art IRL methods by recovering reward functions that leads to optimal, expert-matching behaviors. On policy learning, results demonstrate that policies learned through EAIRL perform comparably to GAIL and AIRL with non-disentangled (state-action) reward function but significantly outperform policies learned through AIRL with disentangled reward (state-only) and GAN interpretation of Guided Cost Learning (GAN-GCL) (Finn et al., 2016a).

## 2 BACKGROUND

We consider a Markov decision process (MDP) represented as a tuple $(\mathcal{S}, \mathcal{A}, \mathcal{P}, \mathcal{R}, \rho_0, \gamma)$ where $\mathcal{S}$ denotes the state-space, $\mathcal{A}$ denotes the action-space, $\mathcal{P}$ represents the transition probability distribution, i.e., $\mathcal{P} : \mathcal{S} \times \mathcal{A} \times \mathcal{S} \to [0, 1]$, $\mathcal{R}(s, a)$ corresponds to the reward function, $\rho_0$ is the initial state distribution $\rho_0 : \mathcal{S} \to \mathbb{R}$, and $\gamma \in (0, 1)$ is the discount factor. Let $q(a|s, s')$ be an inverse model that maps current state $s \in \mathcal{S}$ and next state $s' \in \mathcal{S}$ to a distribution over actions $\mathcal{A}$, i.e., $q : \mathcal{S} \times \mathcal{S} \times \mathcal{A} \to [0, 1]$. Let $\pi$ be a stochastic policy that takes a state and outputs a distribution over actions such that $\pi : \mathcal{S} \times \mathcal{A} \to [0, 1]$. Let $\tau$ and $\tau_E$ denote a set of trajectories, a sequence of state-action pairs $(s_0, a_0, \cdots s_T, a_T)$, generated by a policy $\pi$ and an expert policy $\pi_E$, respectively, where $T$ denotes the terminal time. Finally, let $\Phi(s)$ be a potential function that quantifies a utility of a given state $s \in \mathcal{S}$, i.e., $\Phi : \mathcal{S} \to \mathbb{R}$. In our proposed work, we use an empowerment-based potential function $\Phi(\cdot)$ to regularize policy update under MaxEnt-IRL framework. Therefore, the following

---

[1] Supplementary material is available at `https://sites.google.com/view/eairl`

sections provide a brief background on MaxEnt-IRL, adversarial reward and policy learning, and variational information-maximization approach to learn the empowerment.

## 2.1 MAXENT-IRL

MaxEnt-IRL (Ziebart et al., 2008) models expert demonstrations as Boltzmann distribution using parametrized reward $r_\xi(\tau)$ as an energy function, i.e.,

$$p_\xi(\tau) = \frac{1}{Z} \exp(r_\xi(\tau)) \tag{1}$$

where $r_\xi(\tau) = \sum_{t=0}^{T} r_\xi(s_t, a_t)$ is a commutative reward over given trajectory $\tau$, parameterized by $\xi$, and $Z$ is the partition function. In this framework, the demonstration trajectories are assumed to be sampled from an optimal policy $\pi^*$, therefore, they get the highest likelihood whereas the suboptimal trajectories are less rewarding and hence, are generated with exponentially decaying probability. The main computational challenge in MaxEnt-IRL is to determine $Z$. The initial work in MaxEnt-IRL computed $Z$ using dynamic programming (Ziebart et al., 2008) whereas modern approaches (Finn et al., 2016b;a; Fu et al., 2017) present importance sampling technique to approximate $Z$ under unknown dynamics.

## 2.2 ADVERSARIAL INVERSE REINFORCEMENT LEARNING

This section briefly describes Adversarial Inverse Reinforcement Learning (AIRL) (Fu et al., 2017) algorithm which forms a baseline of our proposed method. AIRL is the current state-of-the-art IRL method that builds on GAIL (Ho & Ermon, 2016), maximum entropy IRL framework (Ziebart et al., 2008) and GAN-GCL, a GAN interpretation of Guided Cost Learning (Finn et al., 2016b;a).

GAIL is a model-free adversarial learning framework, inspired from GANs (Goodfellow et al., 2014), where the policy $\pi$ learns to imitate the expert policy behavior $\pi_E$ by minimizing the Jensen-Shannon divergence between the state-action distributions generated by $\pi$ and the expert state-action distribution by $\pi_E$ through following objective

$$\min_\pi \max_{D \in (0,1)^{S \times A}} \mathbb{E}_\pi[\log D(s, a)] + \mathbb{E}_{\pi_E}[\log(1 - D(s, a))] - \lambda H(\pi) \tag{2}$$

where $D$ is the discriminator that performs the binary classification to distinguish between samples generated by $\pi$ and $\pi_E$, $\lambda$ is a hyper-parameter, and $H(\pi)$ is an entropy regularization term $\mathbb{E}_\pi[\log \pi]$. Note that GAIL does not recover reward; however, Finn et al. (2016a) shows that the discriminator can be modeled as a reward function. Thus AIRL (Fu et al., 2017) presents a formal implementation of (Finn et al., 2016a) and extends GAIL to recover reward along with the policy by imposing a following structure on the discriminator:

$$D_{\xi,\varphi}(s, a, s') = \frac{\exp[f_{\xi,\varphi}(s, a, s')]}{\exp[f_{\xi,\varphi}(s, a, s')] + \pi(a|s)} \tag{3}$$

where $f_{\xi,\varphi}(s, a, s') = r_\xi(s) + \gamma h_\varphi(s') - h_\varphi(s)$ comprises a disentangled reward term $r_\xi(s)$ with training parameters $\xi$, and a shaping term $F = \gamma h_\varphi(s') - h_\varphi(s)$ with training parameters $\varphi$. The entire $D_{\xi,\varphi}(s, a, s')$ is trained as a binary classifier to distinguish between expert demonstrations $\tau_E$ and policy generated demonstrations $\tau$. The policy is trained to maximize the discriminative reward $\hat{r}(s, a, s') = \log(D(s, a, s') - \log(1 - D(s, a, s')))$. Note that the function $F = \gamma h_\varphi(s') - h_\varphi(s)$ consists of free-parameters as no structure is imposed on $h_\varphi(\cdot)$, and as mentioned in (Fu et al., 2017), the reward function $r_\xi(\cdot)$ and function $F$ are tied upto a constant $(\gamma - 1)c$, where $c \in \mathbb{R}$; thus the impact of $F$, the shaping term, on the recovered reward $r$ is quite limited and therefore, the benefits of reward shaping are not fully realized.

## 2.3 EMPOWERMENT AS MAXIMAL MUTUAL INFORMATION

Mutual information (MI), an information-theoretic measure, quantifies the dependency between two random variables. In intrinsically-motivated reinforcement learning, a maximal of mutual information between a sequence of $K$ actions $\boldsymbol{a}$ and the final state $\boldsymbol{s'}$ reached after the execution of $\boldsymbol{a}$,

conditioned on current state $s$ is often used as a measure of internal reward (Mohamed & Rezende, 2015), known as Empowerment $\Phi(s)$, i.e.,

$$\Phi(s) = \max I(a, s'|s) = \max \mathbb{E}_{p(s'|a,s)w(a|s)} \left[ \log \left( \frac{p(a, s'|s)}{w(a|s)p(s'|s)} \right) \right] \tag{4}$$

where $p(s'|a, s)$ is a $K$-step transition probability, $w(a|s)$ is a distribution over $a$, and $p(a, s'|s)$ is a joint-distribution of $K$ actions $a$ and final state $s'^2$. Intuitively, the empowerment $\Phi(s)$ of a state $s$ quantifies an extent to which an agent can influence its future. Thus, maximizing empowerment induces an intrinsic motivation in the agent that enforces it to seek the states that have the highest number of future reachable states.

Empowerment, like value functions, is a potential function that has been previously used in reinforcement learning but its applications were limited to small-scale cases due to computational intractability of MI maximization in higher-dimensional problems. Recently, however, a scalable method (Mohamed & Rezende, 2015) was proposed that learns the empowerment through the more-efficient maximization of variational lower bound, which has been shown to be equivalent to maximizing MI (Agakov, 2004). The lower bound was derived (for complete derivation see Appendix A.1) by representing MI in term of the difference in conditional entropies $H(\cdot)$ and utilizing the non-negativity property of KL-divergence, i.e.,

$$I^w(s) = H(a|s) - H(a|s', s) \geq H(a) + \mathbb{E}_{p(s'|a,s)w_\theta(a|s)}[\log q_\phi(a|s', s)] = I^{w,q}(s) \tag{5}$$

where $H(a|s) = -\mathbb{E}_{w(a|s)}[\log w(a|s)]$, $H(a|s', s) = -\mathbb{E}_{p(s'|a,s)w(a|s)}[\log p(a|s', s)]$, $q_\phi(\cdot)$ is a variational distribution with parameters $\phi$ and $w_\theta(\cdot)$ is a distribution over actions with parameters $\theta$. Finally, the lower bound in Eqn. 5 is maximized under the constraint $H(a|s) < \eta$ (prevents divergence, see (Mohamed & Rezende, 2015)) to compute empowerment as follow:

$$\Phi(s) = \max_{w,q} \mathbb{E}_{p(s'|a,s)w(a|s)}[-\frac{1}{\beta} \log w_\theta(a|s) + \log q_\phi(a|s', s)] \tag{6}$$

where $\beta$ is $\eta$ dependent temperature term.

Mohamed & Rezende (2015) also applied the principles of Expectation-Maximization (EM) (Agakov, 2004) to learn empowerment, i.e., alternatively maximizing Eqn. 6 with respect to $w_\theta(a|s)$ and $q_\phi(a|s', s)$. Given a set of training trajectories $\tau$, the maximization of Eqn. 6 w.r.t $q_\phi(\cdot)$ is shown to be a supervised maximum log-likelihood problem whereas the maximization w.r.t $w_\theta(\cdot)$ is determined through the functional derivative $\partial I/\partial w = 0$ under the constraint $\sum_a w(a|s) = 1$. The optimal $w^*$ that maximizes Eqn. 6 turns out to be $\frac{1}{Z(s)} \exp(\beta \mathbb{E}_{p(s'|s,a)}[\log q_\phi(a|s, s')])$, where $Z(s)$ is a normalization term. Substituting $w^*$ in Eqn. 6 showed that the empowerment $\Phi(s) = \frac{1}{\beta} \log Z(s)$ (for full derivation, see Appendix A.2).

Note that $w^*(a|s)$ is implicitly unnormalized as there is no direct mechanism for sampling actions or computing $Z(s)$. Mohamed & Rezende (2015) introduced an approximation $w^*(a|s) \approx \log \pi(a|s) + \Phi(s)$ where $\pi(a|s)$ is a normalized distribution which leaves the scalar function $\Phi(s)$ to account for the normalization term $\log Z(s)$. Finally, the parameters of policy $\pi$ and scalar function $\Phi$ are optimized by minimizing the discrepancy, $l_I(s, a, s')$, between the two approximations $(\log \pi(a|s) + \Phi(s))$ and $\beta \log q_\phi(a|s', s))$ through either absolute ($p = 1$) or squared error ($p = 2$), i.e.,

$$l_I(s, a, s') = \left| \beta \log q_\phi(a|s', s) - (\log \pi_\theta(a|s) + \Phi_\varphi(s)) \right|^p \tag{7}$$

## 3 EMPOWERED ADVERSARIAL INVERSE REINFORCEMENT LEARNING

We present an inverse reinforcement learning algorithm that learns a robust, transferable reward function and policy from expert demonstrations. Our proposed method comprises (i) an inverse model $q_\phi(a|s', s)$ that takes the current state $s$ and the next state $s'$ to output a distribution over

---

[2] In our proposed work, we consider only immediate step transitions i.e., $K = 1$, hence variables $s$, $a$ and $s'$ will be represented in non-bold notations.

actions $\mathcal{A}$ that resulted in $s$ to $s'$ transition, (ii) a reward $r_\xi(s, a)$, with parameters $\xi$, that is a function of both state and action, (iii) an empowerment-based potential function $\Phi_\varphi(\cdot)$ with parameters $\varphi$ that determines the reward-shaping function $F = \gamma\Phi_\varphi(s') - \Phi_\varphi(s)$ and also regularizes the policy update, and (iv) a policy model $\pi_\theta(a|s)$ that outputs a distribution over actions given the current state $s$. All these models are trained simultaneously based on the objective functions described in the following sections to recover optimal policies and generalizable reward functions concurrently.

### 3.1 INVERSE MODEL $q_\phi(a|s, s')$ OPTIMIZATION

As mentioned in Section 2.3, learning the inverse model $q_\phi(a|s, s')$ is a maximum log-likelihood supervised learning problem. Therefore, given a set of trajectories $\tau \sim \pi$, where a single trajectory is a sequence states and actions, i.e., $\tau_i = \{s_0, a_0, \cdots, s_T, a_T\}_i$, the inverse model $q_\phi(a|s', s)$ is trained to minimize the mean-square error between its predicted action $q(a|s', s)$ and the action $a$ taken according to the generated trajectory $\tau$, i.e.,

$$l_q(s, a, s') = (q_\phi(\cdot|s, s') - a)^2 \tag{8}$$

### 3.2 EMPOWERMENT $\Phi_\varphi(s)$ OPTIMIZATION

Empowerment will be expressed in terms of normalization function $Z(s)$ of optimal $w^*(a|s)$, i.e., $\Phi_\varphi(s) = \dfrac{1}{\beta}\log Z(s)$. Therefore, the estimation of empowerment $\Phi_\varphi(s)$ is approximated by minimizing the loss function $l_I(s, a, s')$, presented in Eqn. 7, w.r.t parameters $\varphi$, and the inputs $(s, a, s')$ are sampled from the policy-generated trajectories $\tau$.

### 3.3 REWARD FUNCTION $r_\xi(s, a)$

To train the reward function, we first compute the discriminator as follow:

$$D_{\xi,\varphi}(s, a, s') = \frac{\exp[r_\xi(s, a) + \gamma\Phi_{\varphi'}(s') - \Phi_\varphi(s)]}{\exp[r_\xi(s, a) + \gamma\Phi_{\varphi'}(s') - \Phi_\varphi(s)] + \pi_\theta(a|s)} \tag{9}$$

where $r_\xi(s, a)$ is the reward function to be learned with parameters $\xi$. We also maintain the target $\varphi'$ and learning $\varphi$ parameters of the empowerment-based potential function. The target parameters $\varphi'$ are a replica of $\varphi$ except that the target parameters $\varphi'$ are updated to learning parameters $\varphi$ after every $n$ training epochs. Note that keeping a stationary target $\Phi_{\varphi'}$ stabilizes the learning as also mentioned in (Mnih et al., 2015). Finally, the discriminator/reward function parameters $\xi$ are trained via binary logistic regression to discriminate between expert $\tau_E$ and generated $\tau$ trajectories, i.e.,

$$\mathbb{E}_\tau[\log D_{\xi,\varphi}(s, a, s')] + \mathbb{E}_{\tau_E}[(1 - \log D_{\xi,\varphi}(s, a, s'))] \tag{10}$$

### 3.4 POLICY OPTIMIZATION POLICY $\pi_\theta(a|s)$

We train our policy $\pi_\theta(a|s)$ to maximize the discriminative reward $\hat{r}(s, a, s') = \log(D(s, a, s') - \log(1 - D(s, a, s')))$ and to minimize the loss function $l_I(s, a, s') = \left|\beta\log q_\phi(a|s, s') - (\log\pi_\theta(a|s) + \Phi_\varphi(s))\right|^p$ which accounts for empowerment regularization. Hence, the overall policy training objective is:

$$\mathbb{E}_\tau[\log\pi_\theta(a|s)\hat{r}(s, a, s')] + \lambda_I\mathbb{E}_\tau[l_I(s, a, s')] \tag{11}$$

where policy parameters $\theta$ are updated using any policy optimization method such as TRPO (Schulman et al., 2015) or an approximated step such as PPO (Schulman et al., 2017).

Algorithm 1 outlines the overall training procedure to train all function approximators simultaneously. Note that the expert samples $\tau_E$ are seen by the discriminator only, whereas all other models are trained using the policy generated samples $\tau$. Furthermore, the discriminating reward $\hat{r}(s, a, s')$ boils down to the following expression (Appendix B.1):

$$\hat{r}(s, a, s') = f(s, a, s') - \log\pi(a|s)$$

where $f(s, a, s') = r_\xi(s, a) + \gamma\Phi_{\varphi'}(s') - \Phi_\varphi(s)$. Thus, an alternative way to express our policy training objective is $\mathbb{E}_\tau[\log\pi_\theta(a|s)r_\pi(s, a, s')]$, where $r_\pi(s, a, s') = \hat{r}(s, a, s') - \lambda_I l_I(s, a, s')$,

---

**Algorithm 1:** Empowerment-based Adversarial Inverse Reinforcement Learning

---

Initialize parameters of policy $\pi_\theta$, and inverse model $q_\phi$
Initialize parameters of target $\Phi_{\varphi'}$ and training $\Phi_\varphi$ empowerment, and reward $r_\xi$ functions
Obtain expert demonstrations $\tau_E$ by running expert policy $\pi_E$
**for** $i \leftarrow 0$ **_to_** $N$ **do**

  Collect trajectories $\tau$ by executing $\pi_\theta$
  Update $\phi_i$ to $\phi_{i+1}$ with the gradient $\mathbb{E}_\tau[\nabla_{\phi_i} l_q(s, a, s')]$
  Update $\varphi_i$ to $\varphi_{i+1}$ with the gradient $\mathbb{E}_\tau[\nabla_{\varphi_i} l_I(s, a, s')]$
  Update $\xi_i$ to $\xi_{i+1}$ with the gradient:

  $$\mathbb{E}_\tau[\nabla_{\xi_i} \log D_{\xi_i,\varphi_{i+1}}(s, a, s')] + \mathbb{E}_{\tau_E}[\nabla_{\xi_i}(1 - \log D_{\xi_i,\varphi_{i+1}}(s, a, s'))]$$

  Update $\theta_i$ to $\theta_{i+1}$ using natural gradient update rule (i.e., TRPO/PPO) with the gradient:

  $$\mathbb{E}_\tau\left[\nabla_{\theta_i} \log \pi_{\theta_i}(a|s)\hat{r}_{\xi_{i+1}}(s, a, s')\right] + \lambda_I \mathbb{E}_\tau\left[\nabla_{\theta_i} l_I(s, a, s')\right]$$

  After every $n$ epochs sync $\varphi'$ with $\varphi$

---

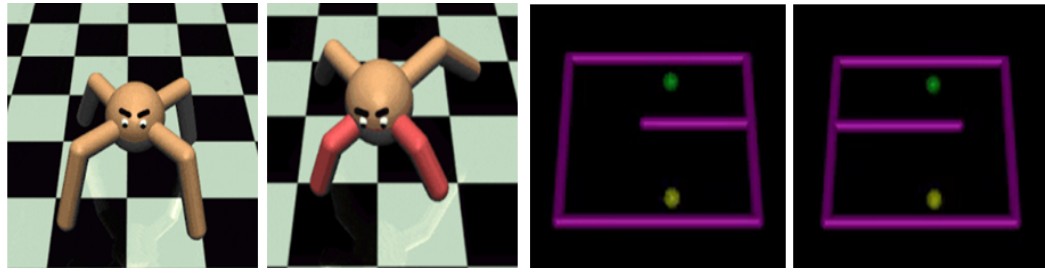

(a) Ant environment        (b) Pointmass-maze environment

Figure 1: Transfer learning problems. Fig. (a) represents a problem where agent dynamics are modified during testing, i.e., a reward learned on a quadruped-ant (left) is transferred to a crippled-ant (right). Fig (b) represents a problem where environment structure is modified during testing, i.e., a reward learned on a maze with left-passage is transferred to a maze with right-passage to the goal (green).

which would undoubtedly yield the same results as Eqn. 11, i.e., maximize the discriminative reward and minimize the loss $l_I$. The analysis of this alternative expression is given in Appendix B to highlight that our policy update rule is equivalent to MaxEnt-IRL policy objective (Finn et al., 2016a) except that it also maximizes the empowerment, i.e.,

$$r_\pi(s, a, s') = r_\xi(s, a, s') + \gamma\Phi(s') + \lambda\hat{H}(\cdot) \tag{12}$$

where, $\lambda$ and $\gamma$ are hyperparameters, and $\hat{H}(\cdot)$ is the entropy-regularization term depending on $\pi(\cdot)$ and $q(\cdot)$. Hence, our policy is regularized by the empowerment which induces generalized behavior rather than locally overfitting to the limited expert demonstrations.

## 4    RESULTS

Our proposed method, EAIRL, learns both reward and policy from expert demonstrations. Thus, for comparison, we evaluate our method against both state-of-the-art policy and reward learning techniques on several control tasks in OpenAI Gym. In case of policy learning, we compare our method against GAIL, GAN-GCL, AIRL with state-only reward, denoted as AIRL($s$), and an augmented version of AIRL we implemented for the purposes of comparison that has state-action reward, denoted as AIRL($s, a$). In reward learning, we only compare our method against AIRL($s$) and AIRL($s, a$) as GAIL does not recover rewards, and GAN-GCL is shown to exhibit inferior performance than AIRL (Fu et al., 2017). Furthermore, in the comparisons, we also include the expert

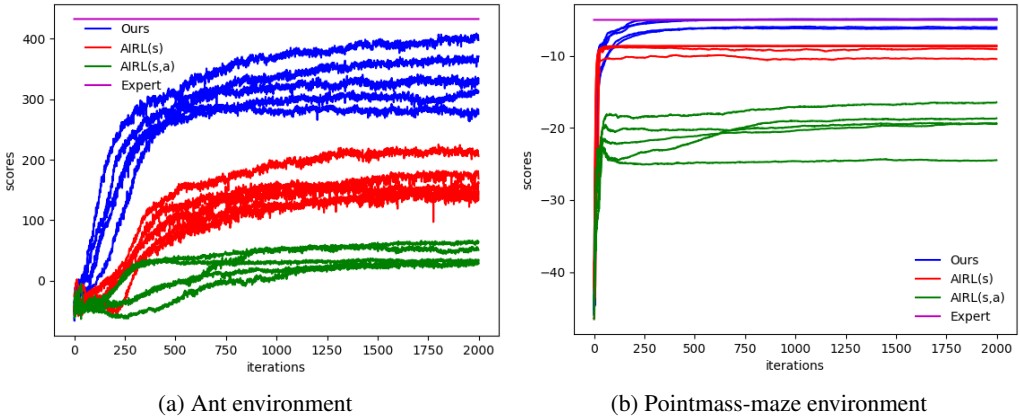

(a) Ant environment          (b) Pointmass-maze environment

Figure 2: The performance of policies obtained from maximizing the learned rewards in the transfer learning problems. It can be seen that our method performs significantly better than AIRL (Fu et al., 2017) and exhibits expert-like performance in all five randomly-seeded trials which imply that our method learns near-optimal, transferable reward functions.

performances which represents a policy learned by optimizing a ground-truth reward using TRPO (Schulman et al., 2015). The performance of different methods are evaluated in term of mean and standard deviation of total rewards accumulated (denoted as score) by an agent during the trial, and for each experiment, we run five randomly-seeded trials.

Table 1: The evaluation of reward learning on transfer learning tasks. Mean scores (higher the better) with standard deviation are presented over 5 trials.

| Algorithm | States-Only | Pointmass-Maze | Crippled-Ant |
|---|---|---|---|
| Expert | N/A | $-4.98 \pm 0.29$ | $432.66 \pm 14.38$ |
| AIRL | Yes | $-8.07 \pm 0.50$ | $175.51 \pm 27.31$ |
| AIRL | No | $-19.28 \pm 2.03$ | $46.12 \pm 14.37$ |
| **EAIRL(Ours)** | **No** | $\mathbf{-7.01 \pm 0.61}$ | $\mathbf{348.43 \pm 43.17}$ |

## 4.1 REWARD LEARNING PERFORMANCE (TRANSFER LEARNING EXPERIMENTS)

To evaluate the learned rewards, we consider a transfer learning problem in which the testing environments are made to be different from the training environments. More precisely, the rewards learned via IRL in the training environments are used to re-optimize a new policy in the testing environment using standard RL. We consider two test cases shown in the Fig. 1.

In the first test case, as shown in Fig. 1(a), we modify the agent itself during testing. We trained a reward function to make a standard quadruped ant to run forward. During testing, we disabled the front two legs (indicated in red) of the ant (crippled-ant), and the learned reward is used to re-optimize the policy to make a crippled-ant move forward. Note that the crippled-ant cannot move sideways (Appendix C.1). Therefore, the agent has to change the gait to run forward. In the second test case, shown in Fig 1(b), we change the environment structure. The agent learns to navigate a 2D point-mass to the goal region in a simple maze. We re-position the maze central-wall during testing so that the agent has to take a different path, compared to the training environment, to reach the target (Appendix C.2).

Fig. 2 compares the policy performance scores over five different trials of EAIRL, AIRL($s$) and AIRL($s, a$) in the aforementioned transfer learning tasks. The expert score is shown as a horizontal line to indicate the standard set by an expert policy. Table 1 summarizes the means and standard deviations of the scores over five trials. It can be seen that our method recovers near-optimal reward functions as the policy scores almost reach the expert scores in all five trials even after transferring to unseen testing environments. Furthermore, our method performs significantly better than both

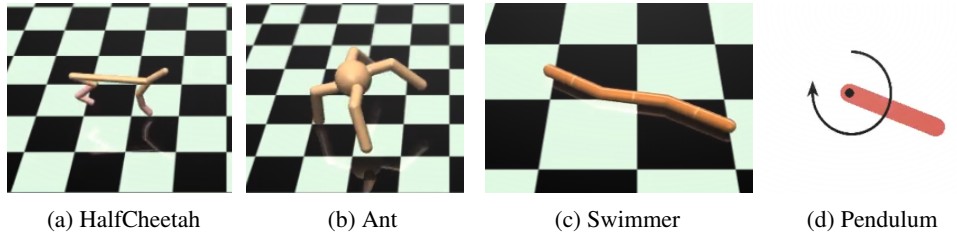

(a) HalfCheetah      (b) Ant      (c) Swimmer      (d) Pendulum

Figure 3: Benchmark control tasks for imitation learning

$\text{AIRL}(s)$ and $\text{AIRL}(s, a)$ in matching an expert's performance, thus showing no downside to the EAIRL approach.

## 4.2 POLICY LEARNING PERFORMANCE (IMITATION LEARNING)

Next, we considered the performance of the learned policy specifically for an imitation learning problem in various control tasks. The tasks, shown in Fig. 3, include (i) making a 2D halfcheetah robot to run forward, (ii) making a 3D quadruped robot (ant) to move forward, (iii) making a 2D swimmer to swim, and (iv) keeping a friction less pendulum to stand vertically up. For each algorithm, we provided 20 expert demonstrations generated by a policy trained on a ground-truth reward using TRPO (Schulman et al., 2015). Table 2 presents the means and standard deviations of policy learning performance scores, over the five different trials. It can be seen that EAIRL, $\text{AIRL}(s, a)$ and GAIL demonstrate similar performance and successfully learn to imitate the expert policy, whereas $\text{AIRL}(s)$ and GAN-GCL fails to recover a policy.

Table 2: The evaluation of imitation learning on benchmark control tasks. Mean scores (higher the better) with standard deviation are presented over 5 trials for each method.

| Methods | Environments | | | |
|---|---|---|---|---|
| | HalfCheetah | Ant | Swimmer | Pendulum |
| Expert | $2139.83 \pm 30.22$ | $935.12 \pm 10.94$ | $76.21 \pm 1.79$ | $-100.11 \pm 1.32$ |
| GAIL | $1880.05 \pm 15.72$ | $738.72 \pm 9.49$ | $50.21 \pm 0.26$ | $-116.01 \pm 5.45$ |
| GCL | $-189.90 \pm 44.42$ | $16.74 \pm 36.59$ | $15.75 \pm 7.32$ | $-578.18 \pm 72.84$ |
| AIRL(s,a) | $1826.26 \pm 19.64$ | $645.90 \pm 41.75$ | $49.52 \pm 0.48$ | $-118.13 \pm 11.33$ |
| AIRL(s) | $121.10 \pm 42.31$ | $271.31 \pm 9.35$ | $33.21 \pm 2.40$ | $-134.82 \pm 10.89$ |
| **EAIRL** | $\mathbf{1870.10 \pm 17.86}$ | $\mathbf{641.12 \pm 25.92}$ | $\mathbf{49.55 \pm 0.29}$ | $\mathbf{-116.26 \pm 8.313}$ |

## 5 DISCUSSION

This section highlights the importance of empowerment-regularized MaxEnt-IRL and modeling rewards as a function of both state and action rather than restricting to state-only formulation on learning rewards and policies from expert demonstrations.

In the scalable MaxEnt-IRL framework (Finn et al., 2016a; Fu et al., 2017), the normalization term is approximated by importance sampling where the importance-sampler/policy is trained to minimize the KL-divergence from the distribution over expert trajectories. However, merely minimizing the divergence between expert demonstrations and policy-generated samples leads to localized policy behavior which hinders learning generalized reward functions. In our proposed work, we regularize the policy update with empowerment i.e., we update our policy to reduce the divergence from expert data distribution as well as to maximize the empowerment (Eqn.12). The proposed regularization prevents premature convergence to local behavior which leads to robust state-action based rewards learning. Furthermore, empowerment quantifies the extent to which an agent can control/influence its environment in the given state. Thus the agent takes an action $a$ on observing a state $s$ such that it has maximum control/influence over the environment upon ending up in the future state $s'$.

Our experimentation also shows the importance of modeling discriminator/reward functions as a function of both state and action in reward and policy learning under GANs framework. The re-

ward learning results show that state-only rewards (AIRL(s)) does not recover the action dependent terms of the ground-truth reward function that penalizes high torques. Therefore, the agent shows aggressive behavior and sometimes flips over after few steps (see the accompanying video), which is also the reason that crippled-ant trained with AIRL's disentangled reward function reaches only the half-way to expert scores as shown in Table 1. Therefore, the reward formulation as a function of both states and actions is crucial to learning action-dependent terms required in most real-world applications, including any autonomous driving, robot locomotion or manipulation task where large torque magnitudes are discouraged or are dangerous. The policy learning results further validate the importance of the state-action reward formulation. Table 2 shows that methods with state-action reward/discriminator formulation can successfully recover expert-like policies. Hence, our empirical results show that it is crucial to model reward/discriminator as a function of state-action as otherwise, adversarial imitation learning fails to learn ground-truth rewards and expert-like policies from expert data.

## 6   CONCLUSIONS AND FUTURE WORK

We present an approach to adversarial reward and policy learning from expert demonstrations by regularizing the maximum-entropy inverse reinforcement learning through empowerment. Our method learns the empowerment through variational information maximization in parallel to learning the reward and policy. We show that our policy is trained to imitate the expert behavior as well to maximize the empowerment of the agent over the environment. The proposed regularization prevents premature convergence to local behavior and leads to a generalized policy that in turn guides the reward-learning process to recover near-optimal reward. We show that our method successfully learns near-optimal rewards, policies, and performs significantly better than state-of-the-art IRL methods in both imitation learning and challenging transfer learning problems. The learned rewards are shown to be transferable to environments that are dynamically or structurally different from training environments.

In our future work, we plan to extend our method to learn rewards and policies from diverse human/expert demonstrations as the proposed method assumes that a single expert generates the training data. Another exciting direction would be to build an algorithm that learns from sub-optimal demonstrations that contains both optimal and non-optimal behaviors.

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

## APPENDICES

## A  VARIATIONAL EMPOWERMENT

For completeness, we present a derivation of presenting mutual information (MI) as variational lower bound and maximization of lower bound to learn empowerment.

### A.1  VARIATIONAL INFORMATION LOWER BOUND

As mentioned in section 2.3, the variational lower bound representation of MI is computed by defining MI as a difference in conditional entropies, and the derivation is formalized as follow.

$$
\begin{aligned}
I^{w,q}(s) &= H(a|s) - H(a|s',s) \\
&= H(a|s) + \mathbb{E}_{p(s'|a,s)w(a|s)}[\log p(a|s',s)] \\
&= H(a|s) + \mathbb{E}_{p(s'|a,s)w(a|s)}[\log \frac{p(a|s',s)q(a|s',s)}{q(a|s',s)}] \\
&= H(a|s) + \mathbb{E}_{p(s'|a,s)w(a|s)}[\log q(a|s',s)] + \mathbb{E}_{p(s'|a,s)w(a|s)}[\log \frac{p(a|s',s)}{q(a|s',s)}] \\
&= H(a|s) + \mathbb{E}_{p(s'|a,s)w(a|s)}[\log q(a|s',s)] + \mathrm{KL}[p(a|s',s)||q(a|s',s)] \\
&\geq H(a|s) + \mathbb{E}_{p(s'|a,s)w(a|s)}[\log q(a|s',s)] \\
&\geq -\mathbb{E}_{w(a|s)} \log w(a|s) + \mathbb{E}_{p(s'|a,s)w(a|s)}[\log q(a|s',s)]
\end{aligned}
$$

### A.2  VARIATIONAL INFORMATION MAXIMIZATION

The empowerment is a maximal of MI and it can be formalized as follow by exploiting the variational lower bound formulation (for details see (Mohamed & Rezende, 2015)).

$$
\Phi(s) = \max_{w,q} \mathbb{E}_{p(s'|a,s)w(a|s)}[-\frac{1}{\beta}\log w(a|s) + \log q(a|s',s)] \tag{13}
$$

As mentioned in section 2.3, given a training trajectories, the maximization of Eqn. 13 w.r.t inverse model $q(a|s',s)$ is a supervised maximum log-likelihood problem. The maximization of Eqn. 13 w.r.t $w(a|s)$ is derived through a functional derivative $\partial I^{w,q}/\partial w = 0$ under the constraint $\sum_a w(a|s) = 1$. For simplicity, we consider discrete state and action spaces, and the derivation is as follow:

$$
\hat{I}^w(s) = \mathbb{E}_{p(s'|a,s)w(a|s)}[-\frac{1}{\beta}\log w(a|s) + \log q(a|s',s)] + \lambda(\sum_a w(a|s) - 1)
$$

$$
= \sum_a \sum_{s'} p(s'|a,s)w(a|s)\{-\frac{1}{\beta}\log w(a|s) + \log q(a|s',s)\} + \lambda(\sum_a w(a|s) - 1)
$$

$$
\frac{\partial \hat{I}^w(s)}{\partial w} = \sum_a \{(\lambda - \beta) - \log w(a|s) + \beta \mathbb{E}_{p(s'|a,s)}[\log q(a|s',s)]\} = 0
$$

$$
w(a|s) = e^{\lambda - \beta} e^{\beta \mathbb{E}_{p(s'|a,s)}[\log q(a|s',s)]}
$$

By using the constraint $\sum_a w(a|s) = 1$, it can be shown that the optimal solution $w^*(a|s) = \frac{1}{Z(s)}\exp(u(s,a))$, where $u(s,a) = \beta \mathbb{E}_{p(s'|a,s)}[\log q(a|s',s)]$ and $Z(s) = \sum_a u(s,a)$. This solution maximizes the lower bound since $\partial^2 I^w(s)/\partial w^2 = -\sum_a \frac{1}{w(a|s)} < 0$.

## B  EMPOWERMENT-REGULARIZED MAXENT-IRL FORMULATION.

In this section we derive the Empowerment-regularized formulation of maximum entropy IRL. Let $\tau$ be a trajectory sampled from expert demonstrations $D$ and $p_\xi(\tau) \propto p(s_0)\Pi_{t=0}^{T-1}p(s_{t+1}|s_t,a_t)\exp^{r_\xi(s_t,a_t)}$ be a distribution over $\tau$. As mentioned in Section 2, the IRL objective is to maximize the likelihood:

$$\max_\xi J(\xi) = \max_\xi \mathbb{E}_D[\log p_\xi(\tau)]$$

Furthermore, as derived in (Fu et al., 2017), the gradient of above equation w.r.t $\xi$ can be written as:

$$\max_\xi J(\xi) = \mathbb{E}_D[\sum_{t=0}^T \frac{\partial}{\partial\xi}r_\xi(s_t,a_t)] - \mathbb{E}_{p_\xi}[\sum_{t=0}^T \frac{\partial}{\partial\xi}r_\xi(s_t,a_t)]$$

$$= \sum_{t=0}^T \mathbb{E}_D[\frac{\partial}{\partial\xi}r_\xi(s_t,a_t)] - \mathbb{E}_{p_{\xi,t}}[\frac{\partial}{\partial\xi}r_\xi(s_t,a_t)]$$

where $r_\xi(\cdot)$ is a parametrized reward to be learned, and $p_{\xi,t} = \int_{s_{t'}\neq t,a_{t'}\neq t} p_\xi(\tau)$ denotes marginalization of state-action at time $t$. Since, it is unfeasible to draw samples from $p_\xi$, Finn et al. (2016a) proposed to train an importance sampling distribution $\mu(\tau)$ whose varience is reduced by defining $\mu(\tau)$ as a mixture of polices, i.e., $\mu(a|s) = \frac{1}{2}(\pi(a|s) + \hat{p}(a|s))$, where $\hat{p}$ is a rough density estimate over demonstrations. Thus the above gradient becomes:

$$\frac{\partial}{\partial\xi}J(\xi) = \sum_{t=0}^T \mathbb{E}_D[\frac{\partial}{\partial\xi}r_\xi(s_t,a_t)] - \mathbb{E}_{\mu_t}[\frac{p_{\xi,t}(s_t,a_t)}{\mu_t(s_t,a_t)}\frac{\partial}{\partial\xi}r_\xi(s_t,a_t)] \tag{14}$$

We train our importance-sampler/policy $\pi$ to maximize the empowerment $\Phi(\cdot)$ for generalization and to reduce divergence from true distribution by minimizing $D_{\mathrm{KL}}(\pi(\tau)\|p_\xi(\tau))$. Since, $\pi(\tau) = p(s_0)\Pi_{t=0}^{T-1}p(s_{t+1}|s_t,a_t)\pi(s_t,a_t)$, the matching terms of $\pi(\tau)$ and $p_\xi(\tau)$ cancel out, resulting into entropy-regularized policy update. Furthermore, as we also include the empowerment $\Phi(\cdot)$ in the policy update to be maximized, hence the overall objective becomes:

$$\max_\pi \mathbb{E}_\pi[\sum_{t=0}^{T-1} r_\xi(s_t,a_t) + \Phi(s_{t+1}) - \log\pi(a_t|s_t)] \tag{15}$$

Our discriminator is trained to minimize cross entropy loss as mention in Eqn. 10, and for the proposed structure of our discriminator Eqn. 9, it can be shown that the discriminator's gradient w.r.t its parameters turns out to be equal to Equation 14 (for more details, see (Fu et al., 2017)). On the other hand, our policy training objective is

$$r_\pi(s,a,s') = \log(D(s,a,s')) - \log(1 - D(s,a,s')) - l_I(s,a,s') \tag{16}$$

In the next section, we show that the above policy training objective is equivalent to Equation 15.

### B.1  POLICY OBJECTIVE

We train our policy to maximize the discriminative reward $\hat{r}(s,a,s') = \log(D(s,a,s') - \log(1 - D(s,a,s')))$ and minimize the information-theoretic loss function $l_I(s,a,s')$. The discriminative reward $\hat{r}(s,a,s')$ simplifies to:

$$\hat{r}(s,a,s') = \log(D(s,a,s')) - \log(1 - D(s,a,s'))$$

$$= \log \frac{e^{f(s,a,s')}}{e^{f(s,a,s')} + \pi(a|s)} - \log \frac{\pi(a|s)}{e^{f(s,a,s')} + \pi(a|s)}$$

$$= f(s,a,s') - \log\pi(a|s)$$

where $f(s,a,s') = r(s,a) + \gamma\Phi(s') - \Phi(s)$. The entropy-regularization is usually scaled by the hyperparameter, let say $\lambda_h \in \mathbb{R}$, thus $\hat{r}(s,a,s') = f(s,a,s') - \lambda_h \log\pi(a|s)$. Hence, assuming

single-sample $(s, a, s')$, absolute-error for $l_I(s, a, s') = |\log q_\phi(a|s, s') - (\log \pi(a|s) + \Phi(s))|$, and $l_i > 0$, the policy is trained to maximize following:

$$r_\pi(s, a, s') = f(s, a, s') - \lambda_h \log \pi(a|s) - l_I(s, a, s')$$
$$= r(s, a) + \gamma\Phi(s') - \Phi(s) - \lambda_h \log \pi(a|s) - \log q(a|s, s') + \log \pi(a|s) + \Phi(s)$$
$$= r(s, a) + \gamma\Phi(s') - \lambda_h \log \pi(a|s) - \log q(a|s, s') + \log \pi(a|s)$$

Note that, the potential function $\Phi(s)$ cancels out and we scale the leftover terms of $l_I$ with a hyperparameter $\lambda_I$. Hence, the above equation becomes:

$$r_\pi(s, a, s') = r(s, a, s') + \gamma\Phi(s') + (\lambda_I - \lambda_h) \log \pi(a|s) - \lambda_I \log q(a|s, s')$$

We combine the log terms together as:

$$r_\pi(s, a, s') = r(s, a) + \lambda_I\Phi(s') + \lambda\hat{H}(\cdot) \tag{17}$$
$$\tag{18}$$

where $\lambda$ is a hyperparameter, and $\hat{H}(\cdot)$ is an entropy regularization term depending on $q(a|s, s')$ and $\pi(a|s)$. Therefore, it can be seen that the Eqn. 17 is equivalent/approximation to Eqn. 15.

## C TRANSFER LEARNING PROBLEMS

### C.1 ANT ENVIRONMENT

The following figures show the difference between the path profiles of standard and crippled Ant. It can be seen that the standard Ant can move sideways whereas the crippled ant has to rotate in order to move forward.

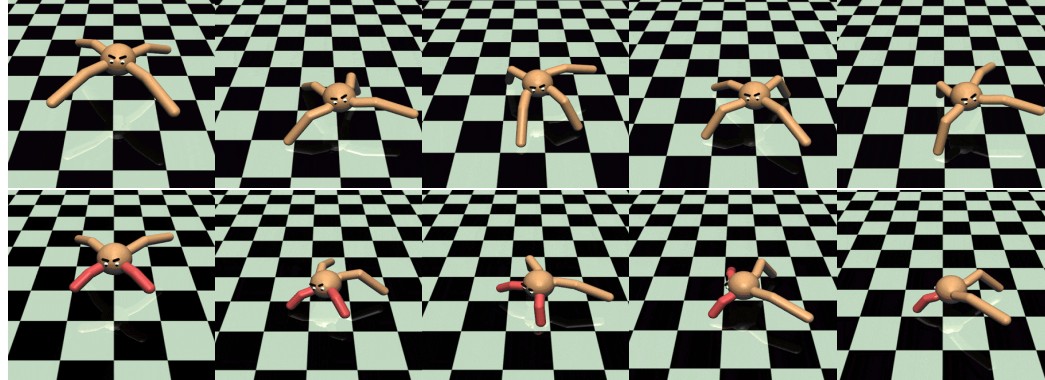

Figure 4: The top and bottom rows show the gait of standard and crippled ant, respectively.

### C.2 MAZE ENVIRONMENT

The following figures show the path profiles of a 2D point-mass agent to reach the target in training and testing environment. It can be seen that in the testing environment the agent has to take the opposite route compared to the training environment to reach the target.

## D IMPLEMENTATION DETAILS

### D.1 NETWORK ARCHITECTURES

We use two-layer ReLU network with 32 units in each layer for the potential function $h_\varphi(\cdot)$ and $\Phi_\varphi(\cdot)$, reward function $r_\xi(\cdot)$, discriminators of GAIL and GAN-GCL. Furthermore, policy $\pi_\theta(\cdot)$ of

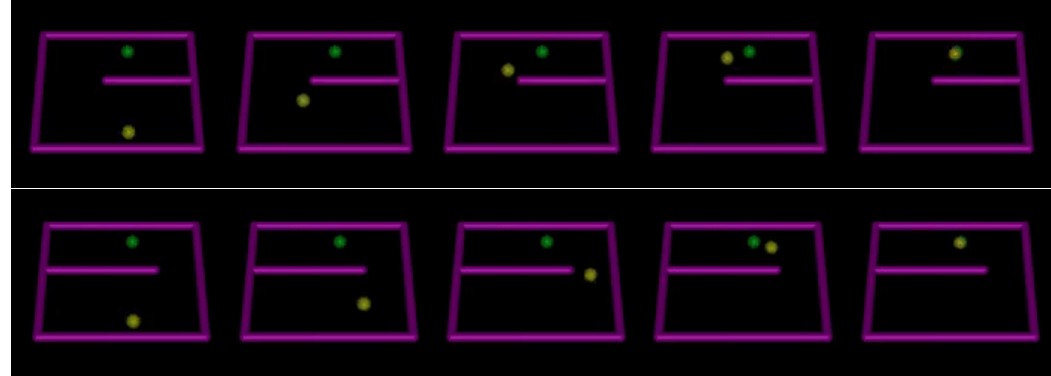

Figure 5: The top and bottom rows show the path followed by a 2D point-mass agent (yellow) to reach the target (green) in training and testing environment, respectively.

all presented models and the inverse model $q_\phi(\cdot)$ of EAIRL are presented by two-layer RELU network with 32 units in each layer, where the network's output parametrizes the Gaussian distribution, i.e., we assume a Gaussian policy.

### D.2 HYPERPARAMETERS

For all experiments, we use the temperature term $\beta = 1$. We evaluated both mean-squared and absolute error forms of $l_I(s, a, s')$ and found that both lead to similar performance in reward and policy learning. We set entropy regularization weight to 0.1 and 0.001 for reward and policy learning, respectively. The hyperparameter $\lambda_I$ was set to 1.0 for reward learning and 0.001 for policy learning. The target parameters of the empowerment-based potential function $\Phi_{\varphi'}(\cdot)$ were updated every 5 and 2 epochs during reward and policy learning respectively. Although reward learning hyperparameters are also applicable to policy learning, we decrease the magnitude of entropy and information regularizers during policy learning to speed up the policy convergence to optimal values. Furthermore, we set the batch size to 2000- and 20000-steps per TRPO update for the pendulum and remaining environments, respectively. For the methods (Fu et al., 2017; Ho & Ermon, 2016) presented for comparison, we use their suggested hyperparameters. We also use policy samples from previous 20 iterations as negative data to train the discriminator of all IRL methods presented in this paper to prevent the parametrized reward functions from overfitting the current policy samples.

