# OpenReview forum: "Adversarial Imitation via Variational Inverse Reinforcement Learning"
_ICLR.cc/2019/Conference_

### Official Review · AnonReviewer2 · 2018-11-01
**Good work, but the important aspects are not discussed**

**Rating:** 6
**Confidence:** 4

**Review:**

The paper proposes a method for inverse reinforcement learning based on AIRL. It's main contribution is that the shaping function is not learned while training the discriminator, but separately as an approximation of the empowerment (maximum mutual information). This shaping term aims to learn disentangled rewards without being restricted to learning state-only reward functions, which is a major restriction of AIRL.

The main weakness of the paper is, that it does not justify or motivate the main deviations compared to AIRL. The new objective for updating the policy is especially problematic because it does no longer correspond to the RL objective but includes an additional term that biases the policy towards actions that increase its empowerment. Although both terms of the update can be derived independently from an IRL and Empowerment perspective respectively, optimizing the sum was not derived from a common problem formulation. By combining these objectives, the learned reward function may lead to policies that fail to match the expert demonstration without such bias. This does not imply that the approach is not sound per se, however, simply presenting such update without any discussion is insufficient--especially given that it constitutes the main novelty of the approach. I think the paper would be much stronger if the update was derived from an empowerment-regularized IRL formulation. And even then, the implications of such bias/regularization would need to be properly discussed and evaluated, in particular with respect to the trade-off lambda, which--again--is hardly mentioned in the submission. I'm also not sure if the story of the paper works out; when we simply want to use empowerment as shaping term, why not use two separate policies for computing the empowerment and reward function respectively. Is the bias in the policy update maybe more important than the shaping term in the discriminator update for learning disentangled rewards?

Keeping these issues aside, I actually like the paper. It tackles the main drawback of AIRL and the idea seems quite nice. Having a reward function that does not actively induce actions that can be explained by empowerment, may not always be appropriate, but often enough it may be a sensible approach to get more generalizable reward functions. The paper is also well written with few typos. The parts that are discussed are clear and the experimental results seem fine as well (although more experiments on the reward transfer would be nice).

Minor notes:
I think there is a sign error in the policy update
Typo in the theorem, grantee should be guarantee

Question:
Please confirm that the reward transfer was learned with a standard RL formulation. Does the learned policy change, when we use the empowerment objective as well?



Update (22.11)
I think that the revised version is much better than the original submission because it now correctly attributes the improved generalization to an inductive bias in the policy update.  However, the submission still seems borderline to me.

- The proposed method uses the empowerment both for regularization as well as for reward shaping, but it is not clear whether the latter improves generalization. If the reward shaping was not necessary, it would be cleaner to use empowerment only for regularization. If the reward shaping is beneficial, this should be shown in an ablative experiment.

- The benefit of using empowerment (whether for reward shaping or for regularization) should be discussed. Empowerment for generalization is currently hardly motivated.

- The derivation could be a bit more rigorous.

As the presentation is now much more sound, I slightly increased my rating.

---

> ### Author Response · Authors · 2018-11-12
> **Response to Reviewer2**
>
> We would like to thank our reviewer for such comprehensive feedback. We have revised the manuscript to address reviewer comments. The response summaries are as follow:
>
> Issue 1: Please confirm that the reward transfer was learned with a standard RL formulation.
> Response:
> Yes, we use standard RL formulation in reward transfer tasks, i.e., the policy is optimized with only the transferred reward and no empowerment bonus.
> Issue 2: Does the learned policy change, when we use the empowerment objective as well?
> Response:
> For the stated values of entropy (λ_h) and information-gain regularizers (λ_I), the policy maximizes the shaped reward and entropy. Shaping rewards induce a policy behavior that leads to learning a generalized reward function. Furthermore, our experiment shows that the policy converges to an expert-like (demonstrated) behavior despite that it maximizes both reward and empowerment.
>
> We include a derivation in the paper to highlight the impact of trade-off lambda on the policy bias towards maximizing the empowerment or imitating the expert behavior. To leave the derivation simple, we have modified the equation (6) to absolute error instead of the mean-square error, and all experimental results are updated accordingly. We have verified that the modification doesn’t impact the results since the purpose of equation (6) is to measure the discrepancy between forward and inverse models. In our paper, we show that the discriminative reward r ̂ simplifies to the following:
>
> r ̂=log⁡[D(s,a,s' )] - log[⁡1-D(s,a,s' )]=f(s,a,s')-λ_h log⁡[π(a│s)]
> ⁡
> The policy is trained to maximize r_π (s,a,s' )=r ̂(s,a,s')-λ_I L_I, that leads to following expression:
>
> r_π (s,a,s' )=f(s,a,s' )+(λ_I-λ_h)log⁡π(a│s)-λ_I log⁡q(a│s,s' )+λ_I Φ(s)
>
> Note that the inverse model q(⋅) is trained using the trajectories generated by the policy π(⋅) (see Algorithm 1) and both models learn distribution over actions. Therefore, maximizing the entropy of q(⋅) is equivalent to maximizing the entropy of π(⋅). Thus, the entropy terms can be combined together as:
>
> r_π (s,a,s' )=f(s,a,s' )+λH(⋅)+λ_I Φ(s)
>
> whereas λ is a function of λ_I and λ_h, and H(⋅) is the entropy. Since, f(s,a,s' )=r(s,a,s' )+γΦ(s' )-Φ(s). The overall policy update rule becomes:
>
> r_π (s,a,s' )=r(s,a,s' )+γΦ(s' )-(1-λ_I)Φ(s)+λH(⋅)
>
> Hence, when λ_h<λ_I<1, the policy objective will be to maximize the shaped reward as well as the entropy. For the stated values of λ_I and λ_h, the policy training is slightly biased toward maximizing the empowerment. The bias of our policy training towards maximizing the empowerment leads to a generalized policy behavior which results in robust reward learning.

---

> > ### Comment · AnonReviewer2 · 2018-11-13
> > **Still too much discrepancy between the text and the algorithm**
> >
> > Thanks for the new derivation. I think it sheds some more light on the policy bias, although I think that setting the inverse model equal to the current policy is going too far and it does not really make sense to talk about "maximizing the entropy of q(.)" given that q is a variational distribution that is fixed during the policy improvement.
> > However, treating the last equation of Appendix B as a rough approximation of the actual objective that is maximized by the policy updates and further assuming that lambda_I=0.99 is close enough to 1, we can see that the policy roughly optimizes "reward + next empowerment". I wonder whether, we could show similar generalization benefits by directly optimizing this objective, e.g. the discriminator could be computed as exp(r+\gamma*\Phi(s'))/(exp(r+\gamma*\Phi(s'))+\pi) and a standard TRPO/PPO update could be used. According to the derivations in Appendix B, this should roughly correspond to the same algorithm. Let's say we can get similar results (potentially replacing \gamma by a hyper-parameter and using higher entropy regularization), such algorithm could be derived in a principled way--from and empowerment-regularized MaxEnt-IRL formulation.
> > In the submitted version, there is a huge discrepancy between the text (generalization is achieved by using empowerment as potential for reward (un)shaping) and the actual algorithm (generalization is achieved by 1% of reward shaping and 99% of policy biasing). These are two completely different approaches; the former does not affect the learned policy (at least in theory) whereas the latter approach relates to regularization and has the potential to lead to much better generalization by preventing overfitting the demonstrations. As these are different approaches, deriving the algorithm from a reward shaping (better: "advantage unshaping") perspective can not be fully sound--which ultimately manifests in the form of a modified policy update rule which is not properly derived. From a reward shaping perspective, the policy for computing the empowerment should not be related at all to the policy that maximizes the reward.
> > I think there is not much missing to turn the submission into a nice paper (if my suggested variant would work out of the box, it might even be possible to revise the submission), however, in the current state the submission is in my opinion not sufficiently sound and almost dangerous, because it gives a wrong impression about the way generalization is achieved.

---

> > > ### Author Response · Authors · 2018-11-15
> > > **Response to Reviewer2: We motivate our work based on Empowerment-regularized MaxEnt-IRL**
> > >
> > > We thank our anonymous reviewer for providing comprehensive and constructive feedback which helped us significantly improve the quality of our paper. We agree with the reviewer, and all modifications have been made to pivot our work around Empowerment-regularized MaxEnt-IRL.
> > >
> > > In the paper (Appendix B), we include the derivation of Empowerment-regularized MaxEnt-IRL. It is highlighted that under empowerment regularization, the policy/importance-sampler is trained to minimize its divergence from the true distribution over expert demonstrations and to maximize the empowerment. The resulting policy update rule (see Eqn. 14 in the paper) becomes:
> > >
> > > max_π⁡ E_π [∑_(t=0)^T r(s,a)+Φ(s' )-log⁡π(a|s)]
> > >
> > > In Appendix B.1, we show that our policy training objective r_π is equivalent to above equation, i.e.,
> > >
> > > r_π (s,a,s' )=log⁡[D(s,a,s' )]-  log[(⁡1-D(s,a,s' )) ]- λ_I L_I      (1)
> > > r_π (s,a,s' )=r(s,a,s' )+γΦ(s' )+λH(⋅)      (2)
> > >
> > > whereas γ and λ are hyperparameters and H(⋅) contains the entropy terms.
> > >
> > > Reviewer’s comment 1: The policy roughly optimizes "reward + next empowerment." I wonder whether we could show similar generalization benefits by directly optimizing this objective.
> > >
> > > Response: We have verified by rerunning the experiments using the above-mentioned simplified policy objective, and it turns out that we obtain the same generalization as obtained by optimizing (1).  Hence, just as our reviewer expected, the empowerment-based regularization prevents the policy from overfitting expert demonstration, thus leads to a generalized behavior which results in learning near-optimal rewards.
> > >
> > > Reviewer’s comment 2: In the submitted version, there is a huge discrepancy between the text and the actual algorithm.
> > >
> > > Response: The discrepancy has been removed. The revised paper now motivates the algorithm based on the notion of Empowerment-regularized MaxEnt-IRL.

---

> > > > ### Comment · AnonReviewer2 · 2018-11-22
> > > > **Story seems to make much more sense now**
> > > >
> > > > Thanks for the revision; I agree that the quality has significantly improved and updated my review.

---

> ### Author Response · Authors · 2018-11-26
> **Response to Reviewer2: Empowerment-based regularization is a consequence of biased reward shaping**
>
> We thank our anonymous reviewer for going through our revised paper and providing us with more constructive feedback. Accordingly, we have further improved our paper especially Section 5 (Discussion) to address our reviewer’s comments.
>
> Reviewer’s comment: The proposed method uses the empowerment both for regularization as well as for reward shaping, but it is not clear whether the latter improves generalization. The benefit of using empowerment (whether for reward shaping or for regularization) should be discussed. Empowerment for generalization is currently hardly motivated.
>
> Response:
>
> We discuss the benefits of using empowerment for regularization as a technique to prevent the policy from overfitting expert demonstrations which leads to learning generalized reward functions. We also present an alternative view of seeing our regularization as a result of biased reward shaping. For more details, please refer to Section 5, paragraph 2-3.
> Summary:
> In the scalable MaxEnt-IRL framework (Finn et.al 2016), the normalization term is approximated by importance sampling where the importance-sampler/policy is trained to minimize the KL-divergence from the distribution over expert trajectories. However, merely minimizing the divergence between expert demonstrations and policy generated samples leads to localized policy behavior which hinders learning generalized reward functions. In our proposed work, we regularize the policy update with empowerment. Hence, we update our policy to reduce the divergence from expert data distribution as well as to maximize the empowerment (Eqn. 12). The proposed regularization prevents premature convergence to local behavior thus leads to robust rewards learning without any restriction on modeling rewards as a function of states only.
> An alternative way to interpret our empowerment-regularized policy optimization is through the perspective of reward shaping. Ng et al. (1999) proposed that the reward shaped with a potential function F of form  γΦ(s' )-Φ(s) does not induce a bias in policy as the optimal policy in MDP M'=(S,A,P,R'=R+F,ρ_0,γ) will also be optimal in the MDP M=(S,A,P,R,ρ_0,γ). However, in our proposed method, we shape our reward R with a discounted empowerment F=γΦ(s') (Eqn. 12) to induce the bias in our policy optimization. The induced bias is due to reward shaping R'=R+F that leads to generalized policy behavior. Furthermore, it is evident that the optimal policy in MDP M'=(S,A,P,R'=R+γΦ,ρ_0,γ) will no longer be optimal in MDP M=(S,A,P,R,ρ_0,γ) as F=γΦ(s') rather than F=γΦ(s' )-Φ(s). However, depending on the hyperparameter γ, the induced bias can be reduced to learn the optimal policies matching the expert behaviors.

---

> > ### Comment · AnonReviewer2 · 2018-12-03
> > **Reward Shaping is not alternate view of regularization**
> >
> > The proposed method uses empowerment both, for reward shaping and for regularization.
> > The reward function is defined as a (properly) shaped reward function, r + \gamma\Phi(s') - \Phi(s) (empowerment-based reward shaping) and the policy is optimized using a regularized TRPO update rule (empowerment-based regularization). Some of the effects _approximately_ cancel out such that the update is similar to a standard TRPO update with reward fuction r + \gamma\Phi(s'). However, this is a quite wild approximation (treating the inverse model as current policy) and hence the actual algorithm uses a combination of reward shaping and regularization. I think it would be much nicer to present a (slighly different) algorithm that only does regularization. This regularization can be achieved by _either_ using a different policy update rule, or (better) by having an additional objective in the reward function. Note that none of these have anything to do with reward shaping!

---

### Official Review · AnonReviewer3 · 2018-11-02
**AR3: Adversarial Imitation via Variational Inverse Reinforcement Learning**

**Rating:** 6
**Confidence:** 3

**Review:**

The authors propose empowerment-based adversarial inverse reinforcement learning (EAIRL), an extension of AIRL which uses empowerment (which quantifies the extent that an agent can influence its state, see eq. 3) as a reward-shaping potential to recover more faithful learned reward functions.

Evaluation:     4/5     Experiments are more preliminary but establish the benefit of the approach.
Clarity:        4/5     Well written. Just a few typos (see below minor comments)
Significance:   4/5     Effective, well motivated approach. Excellent transfer learning results.
Originality:    3.5/5   As the empowerment subroutine is existing work, as is AIRL, combining previous work, but effectively.

Rating:         7/10
Confidence:     3/5     Reviewed this paper in a little less detail than I would prefer, due to time constraints. I will review in more detail and update this and add any additional questions/comments below the minor comments below.

Pros:
- Extension of AIRL which utilizes empowerment to advance the SOE in reward learning
- Well written, related previous work well explained.
Cons:
- Experiments more preliminary
- Combines existing approaches, somewhat incremental

Minor comments:
- grantee (typo), barely utilized -> not fully realized?,

----

Updated review:

After reviewing the comments and the paper in more detail (whose story has evolved substantially) , I have revised my score slightly lower. While in hindsight I can see that the paper has definitely improved, the story has changed rather dramatically, and appears to be still unfolding: the paper's many new elements require further maturation, and that the utility of empowerment for reward shaping and/or regularization to evolve AIRL (i.e. the old story vs. the new story) still needs further investigation/maturation. If the paper is accepted I'm reasonably confident that the authors will be able to "finish up" and address these concerns.
(typo: eq. 4 omits maximizing argument)

---

> ### Author Response · Authors · 2018-11-07
> **Response to reviewer3**
>
> We would like to thank our reviewer for positive feedback.  We would like to satisfy the reviewer concerns about the paper as follow.
> Issue 1: Experiments more preliminary
> Response:
> The transfer learning tasks are challenging. In the case of crippled-ant (see Appendix B.1), the standard ant can move sideways whereas the crippled-ant must rotate to move forward.  Similarly, in-case of point-mass (see Appendix B.2), the agent must take the opposite route compared to training environment to reach the target. These environments test our method for generalizability and ability to learn transferable/portable reward functions.
> ---------
> Issue 2: Combines existing approaches, somewhat incremental
> Response:
> We agree that our method combines the existing approaches. However, the combination is not straightforward, and we combine two approaches in a novel way. In (Mohamed & Rezende 2015), the method uses variational information maximization to learn the empowerment. Once empowerment is determined, it is used as an intrinsic motivation to train a reinforcement learning agent, and the results are presented in simple 2D environments. On the other hand, AIRL learns disentangled reward by restricting state-only reward function, which is a major drawback of their method. Our method uses variational information maximization to learn reward-shaping potential function as empowerment in parallel to learning the reward and policy from expert data, unlike (Mohamed & Rezende 2015) where Empowerment is learned offline. As a result, our method successfully learns portable, near-optimal rewards without being restricted to learning state-only reward functions.  Furthermore, AIRL (Fu et al., 2017),) requires state-only formulation for reward learning and state-action formulation for policy learning whereas our method requires only state-action formulation to learn both rewards and policies from expert demonstrations.

---

### Official Review · AnonReviewer1 · 2018-11-06
**Good empirical results, but overall lacking in clarity with potentially problematic issues**

**Rating:** 6
**Confidence:** 4

**Review:**

Summary/Contribution:
This paper builds on the AIRL framework (Fu et al., 2017) by combining the empowerment maximization objective for optimizing both the policy and reward function. Algorithmically, the main difference is that this introduces the need to optimize a inverse model (q), an empowerment function (Phi) and alters the AIRL updates to the reward function and policy. This paper presents experiments on the original set of AIRL tasks, and shows improved performance on some tasks.

Pros:
    - The approach outperform AIRL by a convincing margin on the crippled ant problem, while obtaining comparable/favorable performance on other benchmarks.

Cons:
    - The justification for using the empowerment maximization framework to learn the shaping parameters is unclear. The formulation introduces a potentially confounding factor by biasing the policy optimization which clouds the experimental picture.

Justification for rating:
This paper presents good empirical results, but without a clear identification of the source of improvement. I lean on the side of rejecting unless the authors can better eliminate any potential bias in their formulation (see question below). The justification for combining the empowerment maximization objective is also unclear while being integral to the novelty of the proposed method.

Questions I could not resolve from my reading:
    - The "imitation learning benchmark" numbers in Table 2 are different from the original AIRL paper. Do the authors have an explanation as to why? Is this only due to a difference in the expert performance?
    - Can the authors confirm that in the transfer experiments, the policy is optimized with only the transfered reward and no empowerment bonus? Otherwise, can the authors comment on whether the performance benefits could be explained by the additional bonus.
    - In equation (12), \Phi is optimized as an (approximate) mutual information, not a value function, so it is not clear why this term approximates the advantage (I suspect this is untrue in EAIRL as V* is recovered at optimality in the AIRL/GAN-GCL formulation). Can the authors comment?
    - Why is w* unnormalized? Unless I am misunderstanding something, in the definition immediately above it, there is a normalization term Z(s).

Other comments:
    - "AIRL(s, a) fails to learn rewards whereas EAIRL recovers the near optimal rewards function" -> This characterization is strange since on some tasks AIRL(s,a) outperforms or is within one standard deviation of EAIRL (e.g. on Half Cheetah, Ant, Swimmer, Pendulum).
    -  "Our experimentation highlights the importance of modeling discriminator/reward functions.. as a function of both state and action". AIRL(s) is better on both the pointmass and crippled-ant task than AIRL(s,a). Can the authors clarify?
    - "Our method leverages .. and therefore learns both reward and policy simultaneously". Can the authors clarify in what sense the reward and policy is being learned simultaneously in EAIRL where it is not in AIRL?
    - In all the tables, the authors' approach is bolded as oppose to the best numbers. I would instead prefer that the authors bold the best numbers to avoid confusion.

- Typos:
    - "the imitation learning methods were proposed"
    - "quantify an extent to which"
    - "GAIL uses Generative Adversarial Networks formulation"
    - "grantee"
    - "no prior work has reported the practical approach"
    - "but, to"
    - "(see (Fu et al., 2017))"

---

> ### Author Response · Authors · 2018-11-07
> **Response to Reviewer1**
>
> We would like to thank our reviewer for such comprehensive reviews.  The response summaries are as follow.
>
> 1: The "imitation learning benchmark" numbers in Table 2 are different from the original AIRL paper. Do the authors have an explanation as to why? Is this only due to a difference in the expert performance?
>
> Response: Yes, the different values are because of the difference in expert performances. For instance, if you notice half-cheetah in our results Table 2, and in AIRL(s,a) (Fu et al., 2017), the results are similar as experts performed comparably.
>
> 2: Can the authors confirm that in the transfer experiments, the policy is optimized with only the transferred reward and no empowerment bonus? Otherwise, can the authors comment on whether the performance benefits could be explained by the additional bonus.
>
> Response: Yes, the policy is optimized using the transferred reward only (no empowerment bonus) using standard reinforcement learning approach.
>
> 3: In equation (12), \Phi is optimized as an (approximate) mutual information, not a value function, so it is not clear why this term approximates the advantage (I suspect this is untrue in EAIRL as V* is recovered at optimality in the AIRL/GAN-GCL formulation). Can the authors comment?
>
> Response: Yes, you are right, equation 12 doesn’t hold for the proposed method.
>
> 4: Why is w* unnormalized? Unless I am misunderstanding something, in the definition immediately above it, there is a normalization term Z(s).
>
> Response: Although w* is defined to be normalized by Z(s), however, there is no direct mechanism for sampling actions or computing Z(s). Therefore, w* is implicitly unnormalized, for more details, please refer to 4.2.2 of ( Mohamed & Rezende 2015).
>
> 5: "AIRL(s, a) fails to learn rewards whereas EAIRL recovers the near optimal rewards function" -> This characterization is strange since on some tasks AIRL(s,a) outperforms or is within one standard deviation of EAIRL (e.g. on Half Cheetah, Ant, Swimmer, Pendulum).
>
> Response: The paper attempts to solve two separate problems, i.e., 1) policy learning and 2) reward learning. For instance, GAIL only solves the policy learning problem and does not recover a reward function. Likewise, AIRL (s, a) can learn a policy (see Table 2) but fails to recover reward function (see Table 1) as it performs poorly on the transfer learning tasks.
>
> 6: Our experimentation highlights the importance of modeling discriminator/reward functions.. as a function of both state and action". AIRL(s) is better on both the pointmass and crippled-ant task than AIRL(s,a). Can the authors clarify?
>
> Response: Please refer to section 5 for details. We highlight the importance of modeling rewards as a function of states and actions in both reward and policy learning problems.
> Policy learning:
> The results show that AIRL with state-only rewards, AIRL(s), fails to learn a policy whereas EAIRL, GAIL, and AIRL that include state-action reward/discriminator formulation successfully recover the policies (see Table 2). Hence, our empirical results show that it is crucial to model reward/discriminator as a function of state-action as otherwise, adversarial imitation learning fails to retrieve policy from expert data.
> Reward learning:
> The results in Table 1 shows that AIRL with state-only rewards (AIRL(s)) does not recover the action dependent terms of the ground-truth reward function that penalizes high torques. Therefore, the agent shows aggressive behavior and flips over after few steps (see the accompanying video). The formulation of rewards as a function of both states and actions is crucial for action regularization in any locomotion or ambulation tasks that discourage actions with large magnitudes. This need for action regularization is well known in optimal control literature and limits the use cases of a state-only reward function in most practical, real-life applications.
>
> 7: "Our method leverages .. and therefore learns both reward and policy simultaneously". Can the authors clarify in what sense the reward and policy is being learned simultaneously in EAIRL where it is not in AIRL?
>
> Response: AIRL with state-action reward formulation (AIRL (s, a)) learns a policy but fails to recover a ground-truth reward function (see Table 1). To determine the reward function, AIRL restricts state-only reward formulation which might be suitable for learning the reward but fails to learn the expert-like behavior policy. Hence, AIRL requires state-only formulation for reward learning and state-action formulation for policy learning whereas our method requires only state-action formulation to learn both rewards and policies from expert demonstrations.
>
> 8: In all the tables, the authors' approach is bolded as opposed to the best numbers. I would instead prefer that the authors bold the best numbers to avoid confusion.
> Response: Modifications made.
>
> 9: Typos
> Response: All the typo errors are removed.

---

> > ### Comment · AnonReviewer1 · 2018-12-09
> > **Question**
> >
> > Are the authors able to release code with this submission?

---

> > > ### Author Response · Authors · 2018-12-09
> > > **EAIRL Code**
> > >
> > > Due to the double-blind submission policy of ICLR, we didn't link the code with our paper, but for now, you can download it here:
> > >
> > > https://drive.google.com/file/d/1wK51y5cERqXgC3H_7Ku5nXEJtKmQB4Rx/view
> > >
> > > Please let me know if you face any trouble downloading/running it.
> > >
> > > Thanks

---

### Author Response · Authors · 2018-11-12
**General response to the reviewers**

We like to thank the anonymous reviewers for their helpful and constructive comments. We provide the individual response to each reviewer's comments. Here we report the list of main changes which we have added to the new revision.

1- We motivate our method through Empowerment-Regularized Maximum Entropy IRL.
2- A discussion on the policy update rule which maximizes both the learned reward function and Empowerment (Appendix B). To leave the derivation simple, we have modified the equation (6) to absolute error instead of the mean-square error, and all experimental results are updated accordingly.
3- Further clarifications on why state-action formulation of reward function is vital to both reward and policy learning (Section 5, Paragraph 3).
4- Further explanations on transfer learning tasks that we use standard RL formulation using only learned rewards, no empowerment to train the agents.
5-Addressed all typological errors mentioned by the reviewers.

---

### Meta-Review · Area_Chair1 · 2018-12-15

**Confidence:** 4
**Recommendation:** Accept (Poster)

**Metareview:**

This paper proposes a regularization for IRL based on empowerment. The paper has some good results, and is generally well-written. The reviewers raised concerns about how the approach was motivated; these concerns have largely been addressed from the reframing of the algorithm from the perspective of regularization. Now, all reviewers agree that the paper is somewhat above the bar for acceptance. Hence, I also recommend accept. There are several changes that the authors are strongly encouraged to incorporate in the final version of the paper (based on discussion between the reviewers):
- The claim that empowerment acts as a regularizer in the policy update is a fairly complicated interpretation of the effect of the algorithm. It relies on an approximation derived in the appendix that relates the proposed objective with an empowerment regularized IRL formulation. The new framing makes much more sense. However, the one sentence reference to this section of the appendix in the main paper is not appropriate given that it is central to the claims of the paper's contribution. More discussion in the main text should be included.
- There are still some parts of the implemented algorithm that could introduce bias (using a target network in the shaping term which differs from the theory in Ng et al. 1999), but this concern could be remedied by a code release. The authors are strongly encouraged to link to the code in the final non-blind submission, especially since IRL implementations tend to be quite difficult to get right.
- The authors said they would change the way they bold their best numbers in their rebuttal. The current paper does not make the promised change, and actually adopts different bolding conventions in different tables which is even more confusing. The numbers should be bolded in a consistent way, bolding the numbers with the best performance up to statistical significance.